# Study on the Mechanism for SIRT1 during the Process of Exercise Improving Depression

**DOI:** 10.3390/brainsci13050719

**Published:** 2023-04-25

**Authors:** Xiao Qiu, Pengcheng Lu, Xinyu Zeng, Shengjie Jin, Xianghe Chen

**Affiliations:** College of Physical Education, Yangzhou University, Yangzhou 225127, China; jsyzqx123@163.com (X.Q.); lpcydty@163.com (P.L.); zxy181101104@163.com (X.Z.); jinshengjie0209@163.com (S.J.)

**Keywords:** depression, SIRT1, exercise, inflammatory factor, gene expression, neurogenesis

## Abstract

The mechanism behind the onset of depression has been the focus of current research in the neuroscience field. Silent information regulator 1 (SIRT1) is a key player in regulating energy metabolism, and it can regulate depression by mediating the inflammatory response (e.g., nuclear factor-kappa B (NF-κB), tumor necrosis factor-α (TNF-α), interleukin-1β (IL-1β)), gene expression in the nucleus accumben (NAc) and CA1 region of the hippocampus (e.g., nescient helix-loop-helix2 (NHLH2), monoamine oxidase (MAO-A), and 5-Hydroxyindole-3-acetic acid (5-HIAA)), and neuronal regeneration in the CA3 region of the hippocampus. Exercise is an important means to improve energy metabolism and depression, but it remains to be established how SIRT1 acts during exercise and improves depression. By induction and analysis, SIRT1 can be activated by exercise and then improve the function of the hypothalamic–pituitary–adrenal (HPA) axis by upregulating brain-derived neurotrophic factors (BDNF), inhibit the inflammatory response (suppression of the NF-κB and TNF-α/indoleamine 2,3-dioxygenase (IDO)/5-Hydroxytryptamine (5-HT) pathways), and promote neurogenesis (activation of the insulin-like growth factor1 (IGF-1) and growth-associated protein-43 (GAP-43) pathways, etc.), thereby improving depression. The present review gives a summary and an outlook based on this finding and makes an analysis, which will provide a new rationale and insight for the mechanism by which exercise improves depression.

## 1. Introduction

Depression is a mental disease with a high incidence, and its mechanism of onset is complex and commonly related to monoamine neurotransmitters, inflammatory factors, dysregulation of the HPA axis, neurotrophic factors, and neural plasticity [1]. SIRT1 is a deacetylase that is widely distributed in the brain. It can affect protein expression via deacetylation of multiple substrates such as NF-κB to play a role in kinds of physiological processes (e.g., cell survival, energy metabolism), and it also functions in depression by protecting the central nervous system (CNS) [2]. Animal experiments revealed that chronic stress led to a reduction in SIRT1 expression in the dentate gyrus (DG) of the hippocampus of mice, inducing depression-like behaviors [3]; injection of SIRT1 activator resveratrol (RSV) in the hippocampus of depressed mice alleviated the depressive symptoms [4]; in addition, infusion of SIRT1 inhibitor, EX-527, into the medial prefrontal cortex of depressed mice reversed the antidepressant effect of S-ketamine [5]. These studies suggest that SIRT1 may exhibit an antidepressant effect (1) by improving the synaptic plasticity of the hippocampus and promoting the production of new neurons, (2) or by regulating the therapeutic effect of antidepressant drugs. While in clinical trials, a blood test of 50 patients with major depression reported that the expression of the SIRT1 gene in peripheral blood was 37% lower than that in the control group [6], and the mutation of the SIRT1 rs10997875 gene locus was positively correlated with the number of patients with depression and the degree of depression [7]. These results indicate that the variation in the SIRT1 gene may have implications for depression, and the expression of the SIRT1 gene is closely related to the onset of depression.

Presently, treatment for depression is still largely confined to pharmacotherapy and psychotherapy. Prior research proved that exercise is an important means to improve depression [8]. It has been reported that 6-week aerobic training increased the content of neurotrophic factors such as BDNF and the volume of the hippocampus, whereas following 6 weeks of detraining, the volume of the hippocampus decreased to baseline [9]. In addition, exercise could also improve the function of the HPA axis, one that is mainly affected by the secretion of glucocorticoids (cortisol) [10], in depressed rats, alleviating the cortisol hypersecretion-induced depression [11]. Moreover, it was found that 4-week swimming reduced the serum level of cortisol in depressed rats, which restored the function of the HPA axis and then alleviated the depression [12]. In a study with 61 depressed college students, the severity of depression and anxiety and the patterns of inflammatory factors, such as TNF-α, interleukin-6 (IL-6), and IL-1β, were determined after 6 weeks of moderate-to-high-intensity interval training, and reductions were demonstrated in depression severity and TNF-α level [13]. This study suggests that moderate-to-high-intensity exercise may improve depression by exhibiting anti-inflammatory effects. Collectively, exercise training can alleviate depression by increasing the volume of the hippocampus, elevating the content of neurotrophic factors such as BDNF, improving the function of the HPA axis, reducing the expression of inflammatory factors, etc. SIRT1 is a key player in regulating depression, and exercise is an important means to improve depression. However, how SIRT1 acts during exercise and then improves depression is yet to be explored. According to the mechanisms of SIRT1 and exercise in improving depression, a speculation is that exercise activates the expression of SIRT1 to improve the HPA axis, inhibiting the inflammatory response and promoting neurogenesis, thereby improving depression. At present, there have been many studies on the improvement of depression by exercise, but there is still a lack of relevant summary on the regulatory role of specific factors in the improvement of depression by exercise. Based on the discussion and analysis of the mechanism of SIRT1 in the occurrence of depression and the improvement of depression by exercise, this article innovatively and systematically demonstrates the specific regulatory mechanism of SIRT1 in the improvement of depression by exercise. It provides a new theoretical basis and idea for the study of exercise improving depression.

## 2. Mechanism of SIRT1 Regulation of Depression

### 2.1. SIRT1 Regulates Depression by Mediating Inflammatory Response

Levels of pro-inflammatory factors such as TNF-α, IL-1β, and IL-6 are elevated in depressed patients. It has been found that the release of TNF-α stimulates the reuptake of 5-HT in rat embryonic raphe cell lines, rapidly activates the serotonin transporter (SERT) through the p38 mitogen-activated protein kinase (p38MAPK)-mediated signaling pathway and reduces the concentration of 5-HT leading to depression [14,15]. In addition, TNF-α significantly upregulates IDO expression by activating inflammatory pathways through the signal transducer and activators of transcription 1 (STAT1), interferon regulatory factor 1 (IRF1), and NF-κB [16]. IDO is the rate-limiting enzyme in the pathway of the 5-HT precursor, tryptophan (TRP), and metabolism toward kynurenine (KYN). Enhancement of IDO activity reduces the content of TRP and promotes the synthesis of KYN, while KYN can be converted to quinolinic acid (QUIN), which is neurotoxic and conducive to depression, in microglial cells [17]. The above studies have shown that TNF-α induces depression via the reuptake of 5-HT, reducing its concentration and activating IDO to produce QUIN.

SIRT1 plays a role in regulating the inflammatory response via various signaling pathways, including the NF-κB and the p38MAPK signaling pathways. Prior research showed that SIRT1 can deacetylate the lysine residue 310 site of the RelA/p65 subunit in NF-κB, inhibit the transcription expression of NF-κB, and reduce the production of TNF-α and other inflammatory factors [18]. Lei et al. [19] found that the expression of IL-1β was upregulated and the mice showed significant depressive symptoms by specifically knocking out SIRT1 in the forebrain microglia of mice. Neuroinflammatory responses regulating depression are closely related to the activation of high-mobility group protein 1 (HMGB1)/toll protein receptor 4 (TLR4)/myeloiddifferentiationfactor88 (MyD88)/NF-κB signaling pathway [20]. RSV is a non-flavonoid organic polyphenol, and it is known as the activator for SIRT1. It was reported that RSV could enhance the activity of SIRT1 in the hippocampus via deacetylation, downregulating the acetylation level of NF-κB and the expression of TNF-α and IL-1β in the hippocampus to suppress the inflammatory response and then improve depressive symptoms [21]. Thus, it is speculated that SIRT1 can directly or indirectly inhibit the activity of NF-κB, thereby alleviating neuroinflammation and then exhibiting the antidepressant effect. In a study where lipopolysaccharides (LPS) induced TNF-α release from macrophages, RSV suppressed TNF-α expression. In addition, this study noted that SIRT1 knock-out upregulated TNF-α expression, which reduced 5-HT concentration and simultaneously activated the activity of IDO, increasing the contents of KYN and QUIN and then inducing depression [22]. The results show that SIRT1 could inhibit the expression of TNF-α. In an animal experiment in mice, 5-week stress reduced the expression of SIRT1, elevated the protein expression of glycogen synthase kinase 3β (GSK3β) in the hippocampal neurons and prefrontal cortex, and increased the number of microglial cells and the level of TNF-α in brain tissue. Following infusion of RSV, the SIRT1 was activated, and the levels of GSK3β protein and TNF-α reversely decreased, contributing to alleviation of depressive symptoms [23]. These studies suggest that SIRT1 plays a role in alleviating depression by decreasing the levels of GSK3β and TNF-α concentration. In recent years, there have also been studies examining the role of SIRT1 in regulating the p38MAPK signaling pathway. The NF-κB proteins exist as p50/p65 dimers under normal circumstances, and their activity is inhibited by inhibitor kappaB (IκB) proteins. Upon activation, the IκB proteins are phosphorylated, ubiquitinated, and degraded by the proteasome and dissociated from the NF-κB proteins. The activated p50/p65 dimers then enter the nucleus to regulate downstream genes including the transcriptional level of inflammatory factors such as TNF-α and IL-6 [24]. Acetylation is an important pathway involved in the function of the NF-κB p65 subunit as a transcription factor. Nuclear translocation of the NF-κB proteins induces acetylation of the p65 subunit by coactivators p300/CBP, resulting in the formation of a transcription complex. Thus, regulation of the acetylation of the NF-κB p65 subunit can affect the gene expression of downstream inflammatory factors, thereby having an effect on the inflammatory response [25]. It was reported that the NF-κB p65 subunit is the direct target of SIRT1 in inflammation, and SIRT1 can reduce the acetylation level of the NF-κB p65 subunit to suppress its effect on the transcription of downstream factors. The NF-κB p65 subunit can be acetylated at many sites with varying functions. It has been proven that the acetylation of lysine at position 310 can enhance the transcription competence of NF-κB [26], while SIRT1 can promote the deacetylation of p65 at a locus at this position. It can be inferred that SIRT1 suppresses the NF-κB signaling pathway by reducing the transcriptional activity of the NF-κB proteins, thereby alleviating depressive symptoms. All these findings suggest that upregulation of SIRT1 inhibits the levels of inflammatory factors such as GSK3β, TNF-α, and NF-κB to suppress the inflammatory response and exhibit anti-inflammatory effects, thereby improving depression-like behaviors.

### 2.2. SIRT1 Regulates Depression by Mediating Related Gene Expression

The incidence of depression is associated with the levels of inflammatory factors, monoamine neurotransmitters, and genes related to the HPA axis. Depression is usually accompanied by disorders in inflammatory factors, which involve multiple pathways such as cytokine–receptor interaction pathways and chemokine signal transduction pathways, suggesting that depression involves the expression of multiple genes involved in inflammation [27]. Inflammatory factors play diverse regulatory roles in different brain regions, disease patterns, and receptor pathways. For instance, IL-6 plays its pro-inflammatory role by potentiating IL-6 trans-signaling through binding soluble IL-6 receptor (sIL-6R) in the CNS; in contrast, IL-6 exhibits an anti-inflammatory effect by potentiating IL-6 signaling via its membrane-bound IL-6 receptor (mIL-6R) [28]. An animal experiment showed that the expression of IL-6 in the CA1 region of the hippocampus decreased in rats under chronic unpredictable mild stress (CUMS). IL-6 knock-out enforced the oxidative stress injury and inflammatory reaction, whereas overexpression of IL-6 suppressed the injury of the nervous system induced by CUMS, promoting neuroprotection and exhibiting an antidepressant effect [29]. Different neurotransmitters act on different cerebral functions. Multiple monoamine neurotransmitters, such as dopamine (DA), norepinephrine (NE), and 5-HT, have shown significance for depression [30]. Previous research found that the IDO level in depressed patients increased, reducing the levels of TRP and subsequent 5-HT and NE. Furthermore, the simultaneous increase in the concentration of the dopamine transporter (DAT) decreased the levels of 5-HT and DA in the synaptic cleft and led to disorders in neurotransmitter metabolism, eventually inducing depression [31]. Moreover, neuroendocrine perturbations involving the HPA axis can also induce depression. Upon stress stimulation, the HPA axis is activated, increasing the levels of corticotropin-releasing hormone (CRH) and subsequent glucocorticoid (GC) in the adrenal cortex. In this case, the intracellular glutamate level increases, leading to poisoning and then cognitive disorders. Meanwhile, the volume of the hippocampus decreases, which reduces the content of corticotropin-releasing factor (CRF), inducing hyperactivation of the HPA axis and then depression-like behaviors [32].

The extracellular signal-regulated kinase (ERK) is involved in the growth, proliferation, and differentiation of neural cells. An animal experiment showed that the ERK pathway in the hippocampus was inhibited upon depression, and ERK regulated BDNF and cAMP response element-binding protein (CREB), two players that are key in regulating depression [33]. A previous study showed that ERK1/2 are potential downstream targets of SIRT1. It was found that CUMS downregulated SIRT1 expression in the hippocampus, resulting in disturbed dendritic architecture and blockage of the phosphorylation of ERK1/2. In contrast, activation of the SIRT1 in the hippocampus increased the level of ERK1/2 phosphorylation following CUMS, restored the dendritic architecture and then exhibited an antidepressant effect [34].

NAc are a group of neurons within the ripples of the hippocampus, and they are responsible for the stimuli that act on the pressure center of the brain. An animal experiment revealed that the SIRT1 mRNA expression increased in the NAc of CUMS mice but decreased in the hippocampus, and there was no significant change in the prefrontal cortex [35]. This study implied that there are differences in the effect of SIRT1 in different brain regions on depression. Kin et al. [36] also found increased expression of SIRT1 mRNA in the NAc of CUMS mice. In addition, the authors noted a further increase following RSV infusion into the NAc, and the mice exhibited more and worse depression-like behaviors. In contrast, infusion of the SIRT1 antagonist, EX-527, into the NAc reduced the severity of depression in the mice. The regulation of depression behavior by SIRT1 in NAc has cell specificity. The emergence of depressive behavior is regulated by the overexpression of dopamine D1 subtype SIRT1 in midspinous neurons, while there is no significant change in SIRT1 expression in dopamine D2 subtype [37]. This suggests that the downregulation of the SIRT1 expression in NAc contributes to the onset of depression.

The upregulation of SIRT1 expression in the CA1 region of the hippocampus causes NHLH2 to deacetylate, increasing the expression of MAO-A. MAO-A can reduce the activity of 5-HT and promote its conversion to 5-HIAA, reducing the contents of 5-HT and NE and then enhancing depression-like behaviors. Following the suppression of SIRT1 expression, the 5-HT homeostasis will be maintained by transcriptional inhibition of MAO-A, improving the synaptic plasticity in the ventral CA1 region. Therefore, suppression of SIRT1 expression can protect neuronal plasticity and maintain the 5-HT homeostasis via the NHLH2/MAO-A pathway, thereby improving depression-like behaviors [38]. Studies have proved that inhibition of SIRT1 can promote the expression of IL-6, and overexpression of SIRT1 in hippocampal CA1 can inhibit IL-6, which is an anti-inflammatory factor with an antidepressant effect in this region [39]. It is presumed that SIRT1 can inhibit the expression of IL-6 in the CA1 region of the hippocampus, thereby inducing depression [40]. Collectively, stress on different brain regions has varying effects on SIRT1 expression, and it may affect SIRT1 expression with different manners and durations. However, the cause of the differences remains to be established.

### 2.3. SIRT1 Regulates Depression by Mediating Neurogenesis

Neurogenesis is a key mechanism in regulating depression. Chronic stress induces perturbed synaptic plasticity of the hippocampus, reduced neurogenesis, and impaired dendritic architecture such as atrophy, eventually causing depression. As reported, SIRT1 can promote the long-term survival of neurons; moreover, it can also protect and improve synaptic plasticity [41]. An animal experiment showed that knock-out of SIRT1 in the hippocampus of mice resulted in memory impairment, reduced secretion of neurons, decreased dendritic complexity, and perturbed synaptic plasticity in the mice, leading to depression-like behaviors [42]. Another study reported that rats under CUMS exhibited reductions in the levels of BDNF, CREB, and SIRT1, increase in miR-134 expression, dendritic remodeling, and DG-CA3 system disorders, while BDNF has a crucial role in maintaining the dendrite length and branching homeostasis [43]. In addition, it was found that infusion of RSV activated SIRT1 expression, promoted CREB phosphorylation, accelerated the transcription of CREB downstream target BDNF in the hippocampus, and downregulated miR-134 expression [44]. Previous research suggested that miR-134, miR-132, and miR-124 play roles in improving the cognitive and memory functions following the onset of depression caused by enhanced synaptic plasticity. Among them, miR-134 was found to be involved in SIRT1-mediated BDNF and CREB expression [45]. miR-134 is a downstream molecule of SIRT1 that can be negatively regulated by SIRT1 [46]. Current studies have noted that SIRT1 plays a role in repairing dendritic architecture and improving depression-like behaviors via the SIRT1/miR-134 signaling in the hippocampus [47]. Furthermore, SIRT1 can upregulate the expression of downstream BDNF, synapsin (SYN), and postsynaptic density protein 95 (PSD95), rescaling the dendritic architecture and restoring the neurogenesis to produce new neurons in the hippocampus [48]. In conclusion, SIRT1 negatively regulates the expression of BDNF, CREB, SYN, and PSD95 through the SIRT1/miR-134 signaling pathway, promoting nerve regeneration and alleviating depression-like behavior.

In addition, other studies have found that the translation process of CREB1 is regulated by miR-124, which negatively regulates CREB expression by binding to the mRNA 3 ‘UTR region of CREB and downregulates the expression of downstream BDNF [49]. It can be inferred that the increase in miR-124 expression in the depressed model can suppress the activity of transcription factor CREB and then decrease the expression of BDNF, inducing a reduction in newly formed neurons and subsequent depression-like behaviors. It was reported that SIRT1 can directly bind to miR-124 and inhibit the expression of miR-124 by deacetylating histone lysine residues [50]. It is presumed that SIRT1 can upregulate the expression of CREB and BDNF and promote neuronal regeneration by inhibiting the expression of miR-124 to repair neurogenesis and improve depressive symptoms. To conclude, SIRT1 can slow depression by modulating neurogenesis (as shown in Table 1).

## 3. Advances in SIRT1 during the Process of Exercise Improving Depression

### 3.1. SIRT1 Gene Expression Pattern during Exercise

Depression occurs with disorders in inflammatory factors (e.g., IL-6), monoamine neurotransmitters (e.g., DA, NE, and 5-HT), and the function of the HPA axis. As reported, exercise has an antidepressant effect, as it can elevate the level of 5-HT by upregulating peroxisome proliferator-activated receptor γ coactiva-tor-1α (PGC-1α), reducing oxidative stress and the inflammatory response (e.g., IL-6 and TNF-α), decreasing IDO activity, and increasing TRY activity [51]. In addition, the expression of PGC-1α-dependent fibronectin type III domain-containing protein 5 (FNDC5) also increases, inducing the expression of BDNF, one that can regulate FNDC5 gene expression [52]. An animal experiment demonstrated that rats under CUMS exhibited reductions in the expression of PGC-1α and FNDC5 in the skeletal muscle and BDNF in the hippocampus, whereas the three factors showed an increasing trend after exercise [53]. Therefore, exercise can ameliorate depression by upregulating PGC-1α, FNDC5, and BDNF expression. Hayek et al. [54] found that exercise can release lactic acid to activate SIRT1 and upregulate the expression of BDNF in the hippocampus through the PGC-1α/FNDC5 pathway, thus playing a neuroprotective role. Thus, it is speculated that exercise could promote antidepressant outcomes by upregulating PGC-1α/FNDC5/BDNF signaling. In conclusion, exercise activation of SIRT1 can upregulate the expression of BDNF in the hippocampus through the PGC-1α/FNDC5 pathway, inhibit the expression of IL-6 and TNF-α, reduce the activity of IDO, promote the expression of 5-HT, improve the central nervous system function, and achieve anti-depression.

miRs are key epigenetic regulators of gene expression and play an important role in the occurrence of depression. A previous experiment reported that SIRT1 inhibited the expression of miR-134 through the repressor complex comprising transcription factor YY1. Additionally, SIRT1 knockdown led to miR-134 overexpression in the CA1 region of the hippocampus in mice, downregulating the expression of CREB and BDNF, resulting in perturbed synaptic plasticity and then depression [55]. This study suggested that SIRT1 may have an antidepressant effect by inhibiting miR-134 and elevating BDNF and CREB levels. SIRT1 can also inhibit the expression of miR-124, miR-134, and miR-138, forming a negative feedback loop to regulate plasticity [56]. In addition, studies found that exercise reduced the expression of the SIRT1 inhibitor, Nicotinamide, in the cerebellum [57], while upregulation of SIRT1 and inhibition of miR-124 expression in the hippocampus enhanced the recovery of mice from stress [58]. In contrast, Liu et al. [59] found that swimming increased the expression of miR-124, miR-134, and miR-138 in the hippocampus of mice under CUMS, improving the depression-like behaviors in the mice. CUMS-induced depression-like behavior was associated with protein kinase B (AKT)/GSK-3β signaling, while swimming significantly increased CREB/BDNF and AKT/GSK-3β signaling in the hippocampus of CUMS mice, and the expression of BDNF was upregulated. Another animal study reported that the expression of CREB and BDNF induced by voluntary running wheel exercise in rats was associated with the increase in the expression of the AKT/GSK-3β signaling in the hippocampus [60]. It is speculated that exercise can downregulate the expression of SIRT1 in a region of the hippocampus, thereby increasing the expression of miR-134. As mentioned before, the SIRT1 expression in the CA1 region of the hippocampus plays a role in promoting depression. Therefore, exercise may downregulate the SIRT1 expression in the CA1 region of the hippocampus to promote the expression of miR-134, activate the CREB/BDNF and AKT/GSK-3β pathways, and upregulate the expression of BDNF, subsequently improving the synaptic plasticity of the hippocampus and eventually exhibiting an antidepressant effect.

Moreover, depression also results in a decrease in the level of glucocorticoid receptor (GR), leading to HPA axis disorders, while administration of antidepressant agents can reduce the peripheral level of miR-124, an upstream regulator of GR. An animal experiment indicated miR-124 overexpression in the brain of depressed rats, but downregulated GR in the hippocampus, and that infusion of miR-124 antagonist upregulated the GR level and then improved depressive symptoms [61,62]. It has been established that exercise can improve the function of the HPA axis by promoting the section of cortisols. Furthermore, exercise can also reverse the changing trend of cortisol and GR in rats under CUMS, promoting the recovery of the HPA axis function to a steady state and then exhibiting an antidepressant effect [63]. Other studies have found that SIRT1 can directly bind to miR-124, promote the formation of heterochromatin by deacetylating histone lysine residues, promote the expression of GR after downregulation of miR-124, and improve the function of the HPA axis [64]. Therefore, it is inferred that exercise can inhibit miR-124 after upregulating SIRT1 expression in the hippocampus, and then upregulate GR to improve HPA axis function and realize anti-depression.

### 3.2. SIRT1 Mediates Inflammatory Response during Exercise

As a key factor in inducing depression, pro-inflammatory factors not only significantly increase the risk of a patient developing depression for the first time, but also, in severe cases, aggravate the patient’s depression-like behavior [65]. There are already studies showing that exercise can improve depression by suppressing the inflammatory response. Liu et al. [66] conducted moderate intensity treadmill training on CUMS mice for 8 weeks, and after aerobic exercise, the expressions of IDO, NF-κB, and TNF-α in depressed mice were significantly downregulated, the inflammatory response decreased, and the expression of 5-HT was significantly upregulated. These results suggest that exercise can significantly inhibit the inflammatory response in depressed mice, and thus play an antidepressant role. The mechanism may be that exercise can inhibit the neuroinflammatory response in the NF-κB and TNF-α/IDO/5-HT signaling pathway and then improve depression. Ventura et al. [67] found that 6 months of aerobic exercise could significantly reduce the level of IL-6, an inflammatory factor associated with depression, and improve depressive symptoms. Zhao et al. [68] also found that serum levels of inflammatory factors IL-6 and interleukin-18 (IL-18) decreased significantly in depressed patients after a total of 18 weeks of cycling exercise intervention. It is suggested that exercise can inhibit proinflammatory factors TNF-α, IL-6, and IL-18 to improve depression. In addition to suppressing proinflammatory factors, exercise can also improve depression by boosting anti-inflammatory factors. Hao [69] confirmed this by showing that hippocampal neurons in depressed rats were damaged, leading to an inflammatory response, while aerobic exercise upregulates the expression of TGF-β1, which has anti-inflammatory effects, protects and repairs the function and structure of hippocampal neurons, and alleviates depression-like behavior in rats. In conclusion, exercise can improve depression by inhibiting inflammatory factors (e.g., TNF-α, IL-6, and IL-18) and promoting anti-inflammatory factor TGF-β1, inhibiting the inflammatory response.

As described above, SIRT1 regulates the NF-κB signaling pathway and reduces the production of inflammatory factors by reducing the transcriptional activity of the NF-κB protein. The above studies confirmed that 8-week moderate intensity exercise can effectively inhibit the hippocampal inflammatory response in mice, and its antidepressant mechanism is that exercise inhibits the neuroinflammatory response in the NF-κB and TNF-α/IDO/5-HT signaling pathway, and SIRT1 also inhibits this inflammatory response and upregulates the expression of 5-HT. Therefore, it is speculated that exercise can upregulate SIRT1 in the hippocampus, thus inhibiting the inflammatory response in the NF-κB and TNF-α/IDO/5-HT pathway, and that it can play an antidepressant effect. Other studies have found that exercise can increase the expression of the SIRT1 protein, reduce the level of the NF-κB protein in the brain, and inhibit neuroinflammation [70,71]. The mechanism is that SIRT1 can directly deacetylate NF-κB at P65 (Lys310) and reduce its overactivation [72]. At the same time, SIRT1 decreased the expression of NF-κB in the brain via other mechanisms. Mee-Inta et al. [73] found that exercise activates nuclear factor erythroid2-related factor2 (Nrf2) after upregulating SIRT1, which further inhibits NF-κB and thus reduces the production of the inflammatory response. In addition, Zhang [74] found that aerobic exercise can improve the expression of the SIRT1 protein to downregulate the mRNA level of NF-κB P65 in TgAPP/PS1 mice, reduce the mRNA levels of proinflammatory factors IL-1β, IL-6, and TNF-α, and increase the anti-inflammatory factors interleukin-10 (IL-10) and tumor necrosis factor-β (TNF-β) mRNA levels, which then inhibits the inflammatory response. Ji et al. [75] also confirmed that anti-resistance exercise can improve the expression level of SIRT1 protein in the hippocampus of insulin-resistant mice to inhibit the levels of NF-κB, NOD-like receptor protein3 (NLRP3) inflammasome, and the IL-1β and IL-18 protein in the hippocampus, which then inhibits the inflammatory response. In conclusion, SIRT1 plays an important regulatory role in the process of exercise ameliorating depression by inhibiting the inflammatory response; that is, exercise can reduce the level of NF-κB protein in the brain, inhibit the pro-inflammatory response of TNF-α, IL-1β, and IL-6, and promote the anti-inflammatory response of IL-10 and TNF-β by upregulating SIRT1. It has an antidepressant effect.

### 3.3. SIRT1 Mediates Neurogenesis during Exercise

Neurogenesis is a process by which new neurons are generated, and it can promote the antidepressant effect. Depression can lead to reductions in the volume of the hippocampus, the density of neurons, the structural complexity of dendritic spines, and neurogenesis in humans [76]. It is reported that exercise can restore or increase the volume of the hippocampus and improve emotional and cognitive functions; however, after cessation of exercise, the volume of the hippocampus will decrease to its original size. This might be due to the fact that exercise upregulates the expression of key proteins, such as the vascular endothelial growth factor (VEGF) and IGF-1, promoting the regeneration of neurons [77]. BDNF is an important antidepressant protein [78]. Exercise can increase the expression of BDNF in the hippocampus and cortex to promote neurogenesis in the hippocampus, during which the proBDNF content increases, resulting in a decrease in the mBDNF/proBDNF ratio but an increase in BDNF mRNA expression [79]. In addition, exercise can affect the functional integrity of the brain via an impact on neural plasticity, and it can also increase the density of axons and neurons and enhance the expression of BDNF and tropomyosin-related kinase receptor in the hippocampus [80]. An animal study demonstrated an increase in the survival rate of neural cells and the percent of new cells differentiated into neurons in depressed rats following 1 week of exercise, and that the BDNF mRNA expression in the hippocampus was much higher relative to the control without exercise [81]. In humans, it was found that aerobic exercise increased the volume of the hippocampus in older adults, and the increase promoted the increase in the serum level of BDNF [82]. In conclusion, exercise can improve depression by upregulating BDNF expression and promoting neurogenesis.

SIRT1 has a high expression in hippocampal neurons and can regulate synaptic plasticity. Related research found that the SIRT1/CREB/BDNF signaling was involved in regulating the cognitive function of rats with obstructive sleep apnea syndrome (OSAS), and 2,3,5,4′-tetrahydroxystilbene-2-O-β-d-glucoside improved the cognitive function and synaptic plasticity by activating the SIRT1/miR-134 pathway and regulating the expression of BDNF and CREB [83]. The evidence available so far indicates that miRs, including miR-134, play roles in regulating the neurogenesis, synaptic plasticity, and neurotransmitter homeostasis in the brain [84]. Combined with the above demonstration that SIRT1/miR-134 can regulate hippocampal plasticity through CREB/BDNF and that exercise can upregulate the expression of BDNF, it is inferred that exercise can promote the increase in BDNF and CREB by improving the expression of SIRT1, thus improving synaptic plasticity, promoting neurogenesis, and relieving depression. Studies revealed that exercise could elevate the levels of IGF-1 and VEGF in peripheral body fluid [85] and promote their penetration through the blood–brain barrier to increase the volumes of the subgranular zone (SGZ) and subventricular zone (SVZ), regulating synaptic plasticity, synaptic density, and mature neurons, eventually reinforcing neurogenesis [86]. Therefore, IGF-1 and VEGF also have critical roles during the process of exercise improving depression. Moreover, the study by Chen et al. [87] reported that exercise strengthened the neuronal plasticity of the hippocampus by upregulating the expression of GAP-43, and IGF-1 was indispensable in the process of GAP-43 regulating neural development and synaptic plasticity. It has been proven that increased expression of IGF-1 can elevate the BDNF level in the hippocampus [88]. According to the above conclusion, SIRT1 can unregulate the expression of SYN and BDNF to promote neurogenesis. Overall, exercise can promote the expression of BDNF and CREB by increasing SIRT1 expression in the hippocampus, and moreover, it can activate the SIRT1/IGF-1/GAP-43 and SYN signaling pathways to increase the volume of SVZ and SGZ, improving neurogenesis, and eventually relieving depression (as shown in Figure 1).

## 4. Conclusions and Prospects

In recent years, under the background of the integration of biology and physical education, there have been a large number of studies on the effect of exercise on improving depression, but there is still a lack of reviews on the regulatory mechanism of specific factors in the process of exercise improving depression. In this regard, based on the fact that SIRT1 is an important factor involved in neuroprotection, and combined with the current popular rehabilitation method of exercise to improve depression, this article conducted an in depth exploration of the mechanism of SIRT1 in depression and the expression of SIRT1 in exercise to improve depression. The conclusions are as follows: (1) Exercise can inhibit the level of NF-κB protein and the pro-inflammatory response of inflammatory factors such as TNF-α, IL-1β, and IL-6 by inducing the increase in SIRT1 expression, and at the same time, along with the upregulation of anti-inflammatory factors such as IL-10 and TNF-β, reduce the brain inflammatory response and improve depression. (2) Exercise upregulates the expression of BDNF in the hippocampus, reduces the activity of IDO, promotes the expression of 5-HT, and improves depression after activating SIRT1 via the PGC-1α/FNDC5 pathway. (3) Exercise can promote BDNF and CREB after increasing SIRT1 expression, activate the SIRT1/IGF-1/GAP-43 and SYN signaling pathway, increase SVZ and SGZ volume, promote neurogenesis, and improve depression. It is worth mentioning that SIRT1 shows different regulatory effects in different brain regions during the regulation of SIRT1 related to miRs expression. Exercise inhibits miR-124 by upregulating SIRT1 expression in the cerebellum and hippocampus, and then upregulates GR to improve the HPA axis function. Exercise promotes miR-134 expression, activates CREB/BDNF and AKT/GSK-3β pathways, and upregulates BDNF expression by downregulating SIRT1 expression in the hippocampal CA1 region. These two regulatory pathways both play a role in improving depression.

However, there are still some problems to be solved and improved in this article, and some prospects for future research are put forward here. This article mainly explored the regulatory mechanism of SIRT1 in the brain, the increase in the expression of SIRT1 in skeletal muscle and myocardial tissue via exercise, and the question of through what pathway SIRT1 acts as an antidepressant. The expression of SIRT1 in the NAc and hippocampus CA1 showed a depression-promoting effect, but whether exercise affects the expression of SIRT1 in NAc remains to be answered. If it does, how does the molecular mechanism behave? Is there any correlation between different exercise items and intensity and the specific effects of SIRT1 expression? This article has not covered everything in detail. The study and elaboration of the above issues will help to further clarify the regulatory mechanism of SIRT1 in exercise-induced depression and will better promote the development of research on exercise-induced depression. At present, the research direction of the mechanism of exercise to improve SIRT1 expression in depression is gradually becoming clear, and exercise will increasingly prove to be an effective intervention and treatment. It is believed that the unknown problems will be revealed one by one in the near future.

## Figures and Tables

**Figure 1 brainsci-13-00719-f001:**
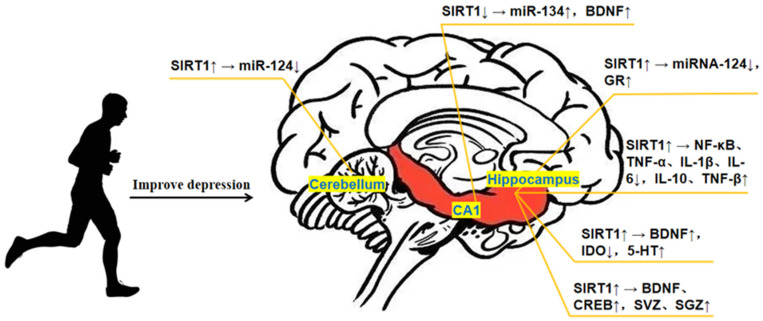
Mechanism of SIRT1 in exercise improving depression.

**Table 1 brainsci-13-00719-t001:** Mechanism of SIRT1 regulating depression.

Scope of Regulation	Functional Changes	Molecular Mechanism
Inflammatory reaction	Hippocampal inflammatory factor↓; Inflammatory reaction↓.	RelA/p65 subunit activity↓; NF-κB deacetylates, Activity and transcription↓; NF-κB p65 acetylation↓; GSK3β↓, TNF-α↓.
Gene expression	Hippocampal dendritic structure recovers; Nucleus accumbens function is damaged; Synaptic plasticity in the ventral CA1 region is damaged.	ERK1/2 phosphorylation↑; NHLH2 deacetylates, MAO-A↑, 5-HT, NE↓; IL-6 in CA1↓.
Neurogenesis	Hippocampal dendritic structure recovers; Synaptic plasticity↑; Neuron function improves.	CREB phosphorylation↑; miR-134↓, miR-124↓; BDNF, CREB, SYN, PSD95↑.

Note: RelA/p65: transcription factor RelA/p65; NF-κB: nuclear factor-kappa B; GSK3β: glycogen synthase kinase 3β; TNF-α: tumor necrosis factor-α; ERK: extra cellular signal-regulated kinase; NHLH2: nescient helix-loop-helix2; MAO-A: monoamine oxidase; 5-HT: 5-Hydroxytryptamine; NE: norepinephrine; IL-6: interleukin-6; CREB: cAMP-response element binding protein; miR: micro RNA; BDNF: brain-derived neurotrophic factor; SYN: synapsin; PSD95: postsynaptic density protein 95.

## Data Availability

Not applicable.

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
