# Peer review of "Study on the Mechanism for SIRT1 during the Process of Exercise Improving Depression"

_brainsci, 2023, doi:10.3390/brainsci13050719_

Round 1
Reviewer 1 Report
I read this manuscript with great interest, but was disappointed to find out that this paper did not add enough to what is already known for it to be sufficiently useful for the readers.
1. As the authors were discussing the mechanism of SIRT1 in depression, specifically in the scenario with exercise, it would be great to provide the information regarding SIRT1 expression levels in depression with a literature summary.
2. There was much overlap regarding the content in part 2 (Mechanism of SIRT1 regulation of depression) and part 3 (Advances in SIRT1 during the process of exercise improving depression).
3. There were lots of subjective assumptions without suffice literature evidence in this review, which I did not believe is appropriate.
Author Response
Dear reviewer, thank you for your valuable review comments. I added the content of SIRT1 expression level in depression and related literature. I deleted the repeated content in Part 1 and Part 2. I added literature evidence to the subjective inference.

Reviewer 2 Report
This review article discussed the mechanism for SIRT1 during the process of exercise improving depression. There are some issues in this manuscript that should be addressed as follows:
- The meaning of the abbreviations in the abstract should be clearly defined at their first mention, e.g.: NHLH2, MAO-A, 5-10 HIAA, NF-kB.
- Page 1 Line 18: The word "make" should be replaced with "makes".
- The novel aspects in this article should be clarified in the Introduction.
- The first sentence in the Introduction "Depression is a mental disease with a high incidence, and its mechanism of onset is complex, commonly related to monoamine neurotransmitters, inflammatory factors, dysregulation of the hypothalamic-pituitary-adrenal (HPA) axis, neurotrophic factors, and neural plasticity" doesn’t have a reference. Please, add one.
- Page 5 Lines 217-219: The sentence "It is presumed that SIRT1 can inhibit the expression of IL-6 in the CA1 region of the hippocampus, thereby inducing depression" doesn’t have a reference. Please, add one.
- Page 6 table 1: A list of abbreviations should be added at the footnote of this table.
- Pages 7 and 8 Lines 327-380: There are only 3 references for about 60 lines. Please add references that explain how SIRT1 mediates the inflammatory response during exercise.
- Lines 326, 381: The word "medicates" should be replaced with "mediates".
- Both figure 1 and table 1 should be cited in the text.
- Conclusion: The clinical value of the data mentioned in this review should be clarified in the conclusion.
- Authors' contribution should be specified.
· General comments:
1. The manuscript should be revised by English-naïve speaker to improve the quality of the language.
2. The manuscript should be checked regarding the grammatical errors and plagiarism.
Author Response
Dear reviewer, thank you for your valuable review comments. I have clearly defined the abbreviations that first appeared in the abstract and corrected the English spelling problems in the article. I modified the other language logic problems in the article. I added references to the content lacking relevant literatures. I explained the novelty of this article in the introduction. I have made a large number of modifications in “SIRT1 mediates inflammatory response during exercise”. I added author contributions and values to the conclusion. I checked the Chinese and English translations carefully.

Round 2
Reviewer 1 Report
The authors addressed some of my concerns.
The authors were discussing the mechanism of SIRT1 in depression, specifically in the scenario with exercise, it would be great to provide a systemic literature summary instead of a few references regarding SIRT1 expression levels in depression.
